# A Facile Graphene Conductive Polymer Paper Based Biosensor for Dopamine, TNF-α, and IL-6 Detection

**DOI:** 10.3390/s23198115

**Published:** 2023-09-27

**Authors:** Md Ashiqur Rahman, Ramendra Kishor Pal, Nazmul Islam, Robert Freeman, Francois Berthiaume, Aaron Mazzeo, Ali Ashraf

**Affiliations:** 1Department of Mechanical Engineering, Purdue University, West Lafayette, IN 47906, USA; rahma144@purdue.edu; 2Hyderabad Campus, Birla Institute of Technology and Science Pilani, Hyderabad 500078, Telangana, India; ramendra.pal@hyderabad.bits-pilani.ac.in; 3Department of Electrical and Computer Engineering, University of Texas Rio Grande Valley, Edinburg, TX 78539, USA; nazmul.islam@utrgv.edu; 4Department of Mechanical Engineering, University of Texas Rio Grande Valley, Edinburg, TX 78539, USA; robert.freeman@utrgv.edu; 5Department of Biomedical Engineering, Rutgers University, Piscataway, NJ 08854, USA; fberthia@soe.rutgers.edu; 6Department of Mechanical & Aerospace Engineering, Rutgers University, Piscataway, NJ 08854, USA

**Keywords:** cancer detection, conductive polymers, dopamine detection, graphene, paper-based biosensor

## Abstract

Paper-based biosensors are a potential paradigm of sensitivity achieved via microporous spreading/microfluidics, simplicity, and affordability. In this paper, we develop decorated paper with graphene and conductive polymer (herein referred to as graphene conductive polymer paper-based sensor or GCPPS) for sensitive detection of biomolecules. Planetary mixing resulted in uniformly dispersed graphene and conductive polymer ink, which was applied to laser-cut Whatman filter paper substrates. Scanning electron microscopy and Raman spectroscopy showed strong attachment of conductive polymer-functionalized graphene to cellulose fibers. The GCPPS detected dopamine and cytokines, such as tumor necrosis factor-alpha (TNF-α), and interleukin 6 (IL-6) in the ranges of 12.5–400 µM, 0.005–50 ng/mL, and 2 pg/mL–2 µg/mL, respectively, using a minute sample volume of 2 µL. The electrodes showed lower detection limits (LODs) of 3.4 µM, 5.97 pg/mL, and 9.55 pg/mL for dopamine, TNF-α, and IL-6 respectively, which are promising for rapid and easy analysis for biomarkers detection. Additionally, these paper-based biosensors were highly selective (no serpin A1 detection with IL-6 antibody) and were able to detect IL-6 antigen in human serum with high sensitivity and hence, the portable, adaptable, point-of-care, quick, minute sample requirement offered by our fabricated biosensor is advantageous to healthcare applications.

## 1. Introduction

Dopamine (DA) is one of the most important neurotransmitters in the human central nervous system. Dopamine deficiency may cause Parkinson’s disease [1], restless leg syndrome [2], schizophrenia [3], and attention deficit hyperactivity disorder (ADHD) [4]. Thus, sensitive and selective dopamine detection using a simple and cost-effective method is highly desirable. On the other hand, cytokines, a class of regulatory proteins, are essential physiological and pathological markers for diagnosis of diseases (e.g., cancer and Alzheimer’s) and response to injury. TNF-α, a pro-inflammatory cytokine originating from macrophages and monocytes, serves as an early sign of inflammatory disease and chronic wounds. The dysregulation of this cytokine can cause various types of diseases including rheumatoid arthritis [5,6], cardiovascular disease [7], Alzheimer’s disease [8], cancer [9,10], and psoriasis [11]. TNF-α is considered an anti-cancer agent, and its receptors can send both survival and death signals to cells. It ranges from 10 pg/mL in healthy human serum to ~2000 pg/mL in patients with chronic wounds [12,13,14,15,16]. As a result, it is crucial to measure TNF-α in a quick, accurate, and simple manner to predict, assess, prevent, and monitor inflammation. Similarly, IL-6 is a multifunctional cytokine that plays an important role in immunomodulation, hematopoiesis, and inflammation. Dysregulated continual synthesis of IL-6 plays a pathological effect on chronic inflammation and autoimmunity including diabetes, rheumatoid arthritis, and cancer [17,18,19,20,21]. Since the concentration of cytokines can be very small, a sensitive sensor is needed to detect them. Electrochemical sensors with microfluidic capability have the potential to detect small quantities of these biomolecules reliably.

Electrochemical biosensors have certain advantages, such as ease of use and rapid, sensitive, and selective response even in complex environments [22,23]. Electrodes for electrochemical sensors include conductive polymers [24,25,26], carbon and metal nanomaterials [27,28,29,30,31,32,33], paper with conductive coating, nanocomposites [34,35,36], ionic liquids [37,38], and other materials. Paper, which is composed of numerous cellulose layers, is a popular substrate due to its low cost, light weight, microporous structure enabling microfluidics, high abundance, environmental friendliness, and ease of bulk manufacture [39]. Thus, paper has drawn much interest in sensor and device fabrication as it is flexible, portable, disposable, and easy to operate [40,41,42,43,44,45,46]. Proper fabrication of electrochemical paper-based biosensors depends on the selection of paper material, design of 2D and 3D electrodes, formation of a hydrophobic wall to delineate microfluidic spreading area, surface modification of electrodes, and analyte conjugation [47]. A popular method of paper surface modification is via the use of conductive polymers.

Conductive polymers such as poly(3,4-ethylenedioxythiophene) (PEDOT) [48], polyaniline (PANI) [49], and polypyrrole (PPy) [50] have been widely used in electrochemical sensing due to their remarkable electrical conductivity, high signal transduction, mechanical flexibility, and chemical stability. Compared to PANI and Ppy, PEDOT shows relatively better stability, excellent charge transport, and higher conductivity. As PEDOT is insoluble in water or in common solvents [51], PEDOT is synthesized in presence of poly(4-styrenesulfonate) (PSS). These two polymers, generally referred to as PEDOT:PSS, exhibit excellent solution–fabrication capability for dip coating. For conductive polymers, an optimized conductivity is needed for a lower signal-to-noise ratio [52]. Graphene, which is an exceptional 2D nanomaterial due to its excellent electrical, mechanical, and thermal properties, can be used as a filler for enhancing PEDOT:PSS conductivity, mechanical property, and high surface area for analyte conjugation [53].

PEDOT-graphene modified electrodes provide a lower dopamine detection limit of 0.33 µM due to low oxidation potential [54]. Aidin et al. developed a sensitive biosensor using semi-conductive poly-(3-thio-phene acetic acid) (P3) to measure TNF-α in human saliva and serum [55]. Wang et al. developed paper-based aptasensors fabricated with conductive polymer nanocomposite electrodes to detect cancer biomarkers [56]. Furthermore, to obtain antigen-antibody conjugation, a graphene screen-printed electrode modified with polyaniline was used and thus exhibited greater surface area for immobilization and exceptional conductivity for human interferon-gamma (IFN-γ) detection [57]. The sensor showed a better response over 5–1000 pg/mL with a detection limit of 3.4 pg/mL. A paper-based biosensor with a carcinoembryonic antigen (CEA) detection range of 6–30 ng mL^−1^ and a detection limit of 2.68 ng mL^−1^ using PEDOT:PSS with modified filter paper was developed by Kumar et al. [58]. On the other hand, Il-6 detection can be carried out via direct [59], indirect competitive [60], and sandwiched nanoparticle-labeled [61,62] electrochemical immunoassay methods. However, the unique combination of graphene-PEDOT:PSS prepared using an optimized recipe and attached to paper-based microfluidic platform via unique processing steps has not been investigated before.

In this paper, we fabricated a miniaturized graphene-PEDOT:PSS coated paper-based biosensor (GCPPS) with laser engraving to detect dopamine, TNF-α, and IL-6. Initially we focused on detectability of the biomolecules using the sensor. For the final test of IL-6 detection in human serum, we triplicated the experiments to verify reproducibility. The electrodes were prepared with planetary mixed conductive G-PEDOT:PSS ink via dip coating, and the performance was analyzed via electrochemical impedance spectroscopy. Towards that goal, we improved the viscosity and stability of the conductive ink by adjusting the ratio of Graphene NanoFlake (GNF) and PEDOT-PSS in DMSO (dimethyl sulfoxide) solvent. Our choice of PEDOT:PSS to act as glue between cellulose fibrils and graphene nanoflakes was able to provide a stable, conductive, high surface area sensing platform. The choice of hydrophilic porous paper substrate, dimethyl sulfoxide (DMSO), as solvent for the diluted ink led to delamination and crust formation prevention. Our results show that Whatman filter paper serves as a microfluidic porous substrate for stable impregnation of conductive and functionalized ink for electrochemical immunoassays. Using brief UV ozone or atmospheric plasma-based dry oxidation, we increased the number of active sites on the sensor surface for antibody attachment via using N-ethyl-N0-(3-(dimethylamino)-propyl)carbodiimide/N-hydroxysuccinimide (EDC/NHS) chemistry. The resultant sensor was able to detect biomolecules such as cytokines in the pg/mL range and in complex environments such as human serum.

## 2. Materials and Methods

### 2.1. Materials

Dopamine hydrochloride, poly(3,4-ethylenedioxythiophene) polystyrene sulfonate (PEDOT: PSS), dimethyl sulfoxide (DMSO), potassium hexacyanoferrate(II) trihydrate, and potassium hexacyanoferrate(III) were obtained from Sigma Aldrich, St. Louis, MO, USA.

Human/Mouse TNF-α Antibody, Recombinant Human TNF-α (HEK293-expressed), Mouse IL-6 Antibody, Recombinant Human IL-6 Protein, and Serpin A1 antigen were purchased from R&D Systems, USA. Human/Mouse TNF-α Antibody, Recombinant Human TNF-α, and Recombinant Human IL-6 were reconstituted at 0.2 mg/mL, 500 µg/mL, and 200 µg/mL in sterile phosphate buffer saline (PBS) respectively. N-Hydroxysuccinimide (NHS) and N-(3-Dimethylaminopropyl)-N′-Ethylcarbodiimide Hydrochloride (EDC) were obtained from Sigma Aldrich, USA. Graphene nanoflakes (surface area 750 m^2^/g, size ~2 µm) were obtained from Sigma Aldrich, USA.

### 2.2. Ink Preparation

The following were added into a container: 1 g of Graphene NanoFlake (GNF), 3 mL of PEDOT:PSS, 3 mL De-Ionized (DI) water, and 800 µL of DMSO solvent (Figure 1a). The solution was then mixed uniformly using a Hauschild Planetary Speed Mixer with rpm 800 and 2500 each for 30 s.

### 2.3. Laser Cutting of Filter Paper and Electrode Dip Coating

Whatman filter paper nos. 1 and 4 were used as a substrate for the electrodes. The process for making the electrodes is shown in Figure 1b. A specific CAD design (biosensor size: L = 13.7 mm, W = 7.2 mm, working electrode diameter = 3.7 mm) was made, and filter papers were cut via laser using 4.8W laser power and 100 pulses per inch (PPI) (Universal Laser System, PLS—4.75). A three-electrode method was used in this experimental design, consisting of counter, working, and reference electrode. The counter electrode (C.E., left one in Figure 1b: Fabricated Sensor) and the working electrode (W.E., middle one in Figure 1b: Fabricated Sensor) were dip-coated in the G-PEDOT:PSS ink, and Ag ink was used for the reference electrode (right one depicted in Figure 1b: Fabricated Sensor). These paper-based electrodes were then dried in vacuum for 24 h. For a stable connection, W.E. and C.E. contact pads were coated with Ag ink. To place the electrodes in a thick paper substrate, double-sided scotch tape was used. After that, hydrophobic PDMS coating was put in between the working zone and contact pads to prevent analyte from flowing through the microfluidic paper from detection zone to contact pads.

### 2.4. Dopamine Solution Preparation

Dopamine hydrochloride was mixed in 1× PBS solvent and a range of dopamine concentrations from 12.5 µM to 400 µM was prepared by serial dilution method.

### 2.5. Surface Modification, EDC/NHS Conjugation, TNF-α Antigen, and IL-6 Antigen Solution Preparation

Initially, the G-PEDOT:PSS electrode surfaces were modified using a Novascan UV-Ozone instrument for 3 min (at 40 °C) to form a carboxylic group (–COOH). Alternatively, similar oxygen functionalization can be obtained using mild oxygen plasma treatment. We performed mild dry plasma oxidation on the sensors at 150 mtorr for 10 s at 20 watts. Then, 0.4 M of EDC was added to the electrode working zone to couple with the -COOH groups and kept in a darker environment for 4 h [63]. Subsequently, 0.2 M NHS was used, and similarly, it was kept in the dark for another 4 h. After that, 20 µL of TNF-α/IL-6 Antibody was added to crosslink with the NHS-treated sensor and kept at 4 °C for 12 h. Recombinant Human TNF-α (antigen)/IL-6 antigen solution of 5 pg/mL to 50 ng/mL was made using the serial dilution method. The detailed schematic of the TNF-α/IL-6 detection protocol is shown in Figure 1c.

### 2.6. Scanning Electron Microscopy (SEM) and Raman Characterization

SEM characterization was carried out on the G-PEDOT:PSS paper-based biosensor (Zeiss, Oberkochen, Germany). Raman spectroscopy, which can reveal signature peaks as well as the number of defects and functionalization from conductive polymers and graphene, was also used. We collected Raman data using a 633 nm laser using a Renishaw inVia reflex system with 50× magnification.

### 2.7. Electrochemical Analysis of Dopamine/TNF-α/IL-6

This paper-based sensor was connected with Autolab PGSTAT302N (from Metrohm) via an adapter to accommodate GCPPS sensors. Cyclic voltammetry (CV) and electrochemical impedance spectroscopy (EIS) were performed to analyze our paper-based electrode sensors. A typical three-electrode method was chosen over the two-electrode method for accurate quantification. The three-electrode method consists of one working electrode, one counter electrode, and one reference electrode in which the potential is kept constant in the working electrode with respect to the reference electrode and the potentiostat measures the current accurately between counter and working electrodes. For dopamine, TNF-α detection, and IL-6 detection, 10 µL of Ferrate (Fe^2+^), 10 µL of Ferric (Fe^3+^) solution (both prepared using PBS solvent), and 2 µL of dopamine/TNF-α antigen/IL-6 antigen was used. To detect IL-6 in human serum solution, 20 µL of human serum and 2 µL of IL-6 antigen were used. In cyclic voltammetry (CV), a current is measured between the working electrode and counter electrode, with the applied potential between the working and reference electrode. The parameters for cyclic voltammetry (CV) were: 0.5 V to 0.5 V with a scan rate of 0.05 V/s. The parameters for impedance spectroscopy were: start frequency—10^6^ Hz, stop frequency—0.1 Hz, frequency per decade—10, DC voltage—10 mV (high DC bias can inhibit precise measurement [64]; therefore, 10 mV constant DC bias was applied in all our tests), AC voltage—10 mV sinusoidal. Analyzing time for each concentration was around 20 min (analyzing time ~5 min and waiting time in between two concentrations ~15 min), total analyzing time from surface modification to first concentration test was around—20.5 h.

## 3. Results and Discussion

### 3.1. Characterization

The enhanced affinity of conductive polymer (CP) to cellulose fibers and interaction of CP with GNF allows GNF to be glued to the fibers. PEDOT:PSS self-assembles on the fibril in the wet stage and becomes π-stacked after drying and, therefore, act as a glue to connect graphene with cellulose fibers [65]. This can be observed from the SEM images of the GNF-CP paper-based sensor in Figure 2a,b. The flat GNF sheets are connected to the fibrous structure of cellulose. The hierarchical structure with more prominent pores and fibers can be observed in low-magnification SEM images in Appendix A. The choice of ink components, mixing technique, drying conditions, and substrate are critical for a stable and effective sensor. We observed film delamination on hard ITO and hydrophobic Fabriano paper substrates and under rapid drying conditions (Appendix A). We also tried Cyrene as a solvent for GNF-CP instead of DMSO. However, the film’s conductivity with Cyrene is low compared to film prepared with DMSO (Appendix A). A Flacktek mixer was used to achieve uniform mixing and introduce curvature to GNF. The dual asymmetric centrifuge of the mixer ensures uniform mixing and removal of bubbles and introduces curvature-induced strain to the GNF and π-π* interaction of GNF with PEDOT:PSS, leading to enhancement of charge carrier concentration of GNF (doping) [66,67,68]. We have also experimented with more viscous ink (containing double the amount of PEDOT:PSS compared to the original ink), which leads to the formation of a thick crust on the surface, preventing access to the micro–nano porous microfluidic structure underneath (Appendix A).

Successful oxidation can be verified via Raman spectroscopy, which can show signature peaks from CP and graphene, and additionally, the number of defects and functionalization (D band). This mild oxidation treatment leads to broadening of the D peak (~1350 cm^−1^), as can be observed from before and after oxidation treatment in the Raman spectra (Figure 2c,d). The G band (~1580 cm^−1^) and 2D band (~2690 cm^−1^) intensity ratio also changed due to the mild oxidation treatment (I(2D/G) from 0.33 to 0.5). This can be due to partial etching of multilayer graphene and formation of -COOH functional group at the edge or defect sites generated by the dry oxidation treatment.

### 3.2. Dopamine Detection

The fabricated G-PEDOT:PSS sensor was employed to investigate the electrochemical detection of dopamine. The dopamine detection was performed using both CV and EIS (Figure 3 and Appendix Aa,b). The negative peak current (reduction) for dopamine showed an increase with the concentration (25 µM to 400 µM) (Appendix A).

Nyquist plots (imaginary impedance vs. real impedance) are typically utilized to investigate interface properties. The lower frequency corresponds to the diffusion-limited process and the high frequency represents the charge transfer resistance [69]. The charge transfer resistance can be found from the diameter of the semicircle formed. Using Nyquist plots is advantageous for characterizing biosensing performance because of its high sensitivity towards detecting the biomolecules [70,71].

The charge-transfer resistance (R_ct_) at the electrode surface depends on the total analytes on the electrode working zone. The charge transfer resistance (semi-circle diameter) was the same for the two similar runs (Appendix A), showing that the sensor surface was stable during electrochemical testing. EIS analysis of dip-coated electrodes (both counter and working electrodes are dip-coated with graphene-conductive ink, reference electrode is dip-coated with Ag) for dopamine detection is shown in Figure 3a,b. The charge transfer resistance increased with the dopamine concentration from 12.5 µM to 400 µM due to greater dopamine molecule adsorption on the surface (Figure 3c). Above 400 µM, a dopamine concentration of 800 µM was also tested, and the Nyquist plot is shown in Appendix A. However, the charge transfer resistance decreased when the concentration increased from 400 µM to 800 µM, and this may be attributed to the working electrode area reaching saturation. This means the maximum number of biomolecules had attached to the surface, and the remaining biomolecules going back in electrolyte therefore decreased the charge transfer resistance [72,73,74]. The Nyquist plot obtained in Figure 3a,b was fitted with Randles equivalent circuit (4 components: solution resistance (R_s_), charge transfer resistance (R_ct_), double-layer capacitance (C_dl_), and Warburg diffusion (Z_w_)) as shown in Figure 3d.

To improve electrical conductivity and detection sensitivity, we dip-coated the sensor a second time using G-PEDOT:PSS ink. Figure 4a,b shows the Nyquist plot for double-layer dip-coated electrodes for the same dopamine concentration range, and Figure 4c indicates the corresponding charge transfer resistance increasing with dopamine concentration from 12.5 µM to 400 µM. The R_ct_ values for double-layer dip-coated electrodes (4.75 kΩ) were less than the single-layer dip-coated ones (20.1 kΩ), which indicates higher electrical conductivity for charge transport. The detection limit of dopamine can be calculated using the formula LOD = $\frac{\sigma }{S}\;$(*σ* is the standard deviation of the response and *S* is the slope = 0.13 × (slope of semi-log plot)) [75]. The LOD for dopamine detection was ~3.4 µM (calibration curve in Figure 4d). The limit of quantification (LOQ) is also similar to the LOD, the lowest level of analyte that can be detected within a certain degree of uncertainty (by considering a signal to noise ratio of 10:1). Thus, the LOQ can be also obtained from the limit of detection (LOD) using the relation: LOQ = 3.3*LOD [76]. For our paper-based biosensor, the LOQ for dopamine detection was ~11.22 µM.

Our paper-based sensors demonstrated a similar or higher sensitivity and range for dopamine detection compared to prior work (Table 1). For example, Yang et al. fabricated AuNPs@PANI nanocomposites to detect dopamine (range: 10–1700 µM, LOD: 5 µM) and ascorbic acid simultaneously [77]. Au nanoparticles on a polyaniline-modified electrode surface for the detection of dopamine (range: 20–100 µM, LOD: 16 µM) was fabricated by Mahalakshmi et al. [78]. PEDOT:PSS organic electrochemical transistor (range: 5–100 µM, LOD: 6 µM) [79], commercial screen-printed electrode modified by PEDOT:PSS/Chitosan/Graphene (range: 0.05–70 µM, LOD: 0.29 µM, more sensitive but with a smaller detection range) [80], and multi-walled carbon nanotube (MWCNT)-PEDOT (range: 10–330 µM, LOD: 10 µM) [81] sensors were the PEDOT-based biosensors reported for dopamine detection.

### 3.3. TNF-α Detection

Similar to dopamine detection, EIS was performed to quantify the TNF-α detection using a Nyquist plot in a double-layer dip-coated sensor. The G-PEDOT:PSS electrode surface was first modified using UV–ozone treatment, and the Nyquist plot before and after the UV–ozone treatment is shown in Appendix A. It indicates that the surface underwent modification after UV–ozone as the resistance value increased. Figure 5a,b refers to the Nyquist plot for TNF-α detection in the concentration range from 5 pg/mL to 50 ng/mL (the same equivalent circuit fit was used, as depicted in Figure 3d). R_ct_ with its logarithmic *x*-axis (concentration) plot is presented in Figure 5c. The plot shows an almost linear straight line with a slope of 66.89. The limit of detection for TNF-α was ~5.97 pg/mL using the formula LOD = $\frac{\sigma }{S}\;$(*σ* is the standard deviation of the response and *S* is the slope = 0.13×(slope of semi-log plot)). The LOQ for TNF-α detection was 19.7 pg/mL.

The performance comparison of different electrochemical biosensors including ours is shown in Table 2. Researchers adopted different structures and techniques for TNF-α detection. Poly(guanine)-functionalized silica NP biosensors showed a detection range of 0.1–100 ng/mL and an LOD of 50 pg/mL [86]. Yin et al. developed alkaline phosphatase-functionalized nanosphere-based biosensors for TNF-α detection (range: 0.02–200.00 ng/mL, LOD: 0.01 ng/mL) [87]. A Au working electrode (range: 10–100 ng/mL, LOD: 10 ng/mL) [88] and comb-structured Au microelectrode array (range: 0.001–1 ng/mL, LOD: 1 pg/mL) [89] -based biosensors were also employed for TNF-α detection. Thus, our sensor shows sensitivity similar to the Au electrode (fabrication of Au electrodes requires complicated processing and bears a high cost).

### 3.4. IL-6 Detection

For IL-6 detection in a wide range (0.002–2000 ng/mL), EIS was performed in a similar fashion to TNF-α. Figure 6a,b shows the Nyquist plot for IL-6 detection (the same equivalent circuit fit was used, as depicted in Figure 3d). Charge transfer resistance (R_ct_) increases with IL-6 concentration (Figure 6c). Similar to dopamine and TNF-α detection, the limit of detection of our paper-based biosensor was calculated using the formula, LOD = $\frac{3.3\sigma }{S}$, and LOD and LOQ were ~9.55 pg/mL, and ~31.5 pg/mL respectively. Previously, Russel et al. developed a Au-based needle-shaped microelectrode for IL-6 detection (range: 20–100 pg/mL, LOD: 20 pg/mL) [59]. Graphene oxide (GO)-based liquid-gated FET biosensors were also used to detect IL-6 in a range of 4.7–300 pg/mL (LOD: 1.53 pg/mL) [93]. In comparison to our paper-based biosensor, the above-mentioned microelectrode/FET needs a complex and costly manufacturing process to obtain almost similar sensitivity.

### 3.5. Selective Detection of IL-6

With the specific antibody attached to the biosensor, the specific antigen can bind with the antibody. Otherwise, using a different antigen will not change the R_ct_. For the selective detection test, the paper-based sensor was initially conjugated with IL-6 antibody. Thus, the signal or charge transfer resistance will change when the IL-6 antibody conjugates only with the IL-6 antigen. Firstly, controlled PBS and IL-6 were added, and there was a significant shift from PBS signal to IL-6 due to antigen attachment. However, there was no significant charge transfer resistance change after Serpin A1 addition, which means no Serpin A1 attachment to the IL-6 antibody (Figure 6d). Adding up to 6 µL of Serpin A1 antigen, no resistance change was observed compared to IL-6, thus signifying that our paper-based biosensor is highly selective. An additional plot for selective detection is shown in Appendix A; the charge transfer resistance did not change significantly after the addition of Serpin A1. Hence, this proves the highly selective detection of antigen/antibody via our paper-based biosensor.

### 3.6. IL-6 Detection with Human Serum

Figure 7 shows that the as-produced G-PEDOT:PSS biosensor provides a nearly linear relation for IL-6 detection in human serum from the concentration 2 pg/mL to 200 ng/mL. The Nyquist plot shown in Figure 7a indicates that the charge transfer resistance, Rct (semi-circle diameter), increased with the concentration (due to more IL-6 antigen molecules becoming attached to the working electrode with increasing concentration and acting as a barrier for charge transfer). Each of the concentrations for IL-6 was replicated 3 times to obtain the uncertainty related to the specific concentration, and Figure 7b shows that each of the IL-6 concentrations maintained the charge transfer resistance within an acceptable range and Rct for no two concentrations overlap each other. The reason behind this linearity and selectivity could be ascribed to the large surface area of the graphene-based working electrode due to graphene nanomaterial and strong covalent bonding between IL-6 antibody and graphene. Additionally, the abundant active sites on graphene for antibodies attached after UV–ozone treatment allows for detection of a wide range of concentrations spanning from pg/mL to ng/ML. The complex environment of human serum did not hinder the detectability of this sensor, and the performance was similar to what was observed in PBS solution (Figure 6). In a healthy subject, the average range of Il-6 is 4.631–5.740 pg/mL [94], whereas the average cut-off Il-6 range for different cancer patients is 1.9–130 pg/mL [95]. As our sensor is capable of detecting the Il-6 concentrations mentioned in the above range, the sensor can be calibrated for early-stage cancer detection.

## 4. Conclusions

Paper-based microfluidic biosensors have shown a great deal of promise in recent years due to their simplicity, high sensitivity, porosity, ease of use, portability, accessibility, and low cost. Furthermore, these sensors can provide data quickly without the need for a laboratory or a skilled individual. One of the key issues to fabricate and functionalize highly sensitive nanomaterials such as graphene to paper substrate has been addressed in our work. In this study, a laser-cut, miniaturized, paper-based electrochemical sensor coated with G-PEDOT:PSS (GCPPS) ink, prepared using a planetary mixer, was developed. Unique planetary mixing with an optimized recipe led to a uniform, stable ink coating of porous paper substrates. Additionally, an innovative dry oxidation step was utilized to functionalize and strongly attach antibodies to the sensor surface. The developed sensor demonstrated excellent performance in detecting dopamine, TNF-α, and IL-6 after modification via mild dry oxidation. The detection range for dopamine, TNF-α, and IL-6 was 12.5–400 µM, 0.005–50 ng/mL, and 0.002–2000 ng/mL, respectively. The sensor also demonstrated a linear relationship between charge transfer resistance and IL-6 concentration in human serum. Additionally, the detection limits were 3.4 µM, 5.97 pg/mL, and 9.55 pg/mL, respectively, indicating that our paper-based biosensor might be a promising avenue for early cancer detection, chronic wound monitoring, or immune sensing. Moreover, the sensor was able to function in human serum, mimicking real-world testing scenarios. Thus, these paper-based sensors offer an alternate, excellent, and reliable platform for immune sensing or early diagnosis even in countries with limited resources. In future, we will investigate the performance of the paper-based biosensor prepared using additive manufacturing techniques for improved electrode fabrication efficiency.

## 5. Patents

This work resulted in the following patent applications.

Ashraf, A., Mazzeo, A.D., Pal, R.K. and Berthiaume, F., Rutgers State University of New Jersey, 2023. Graphene-conductive polymer-coated, paper-based nano-biosensor for cytokine detection. U.S. Patent Application 17/822,641.

## Figures and Tables

**Figure 1 sensors-23-08115-f001:**
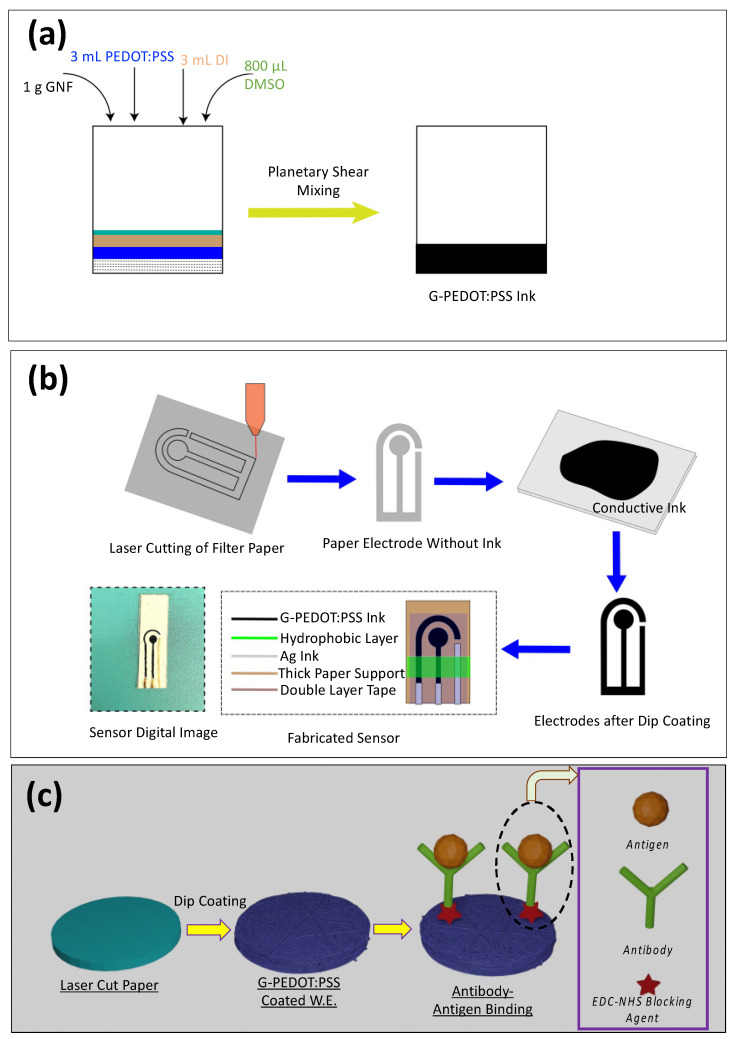
Schematic of (**a**) G-PEDOT:PSS conductive polymer ink preparation. (**b**) Paper-based biosensor fabrication. (**c**) Schematic representation of TNF-α/IL-6 detection protocol for paper-based electrodes.

**Figure 2 sensors-23-08115-f002:**
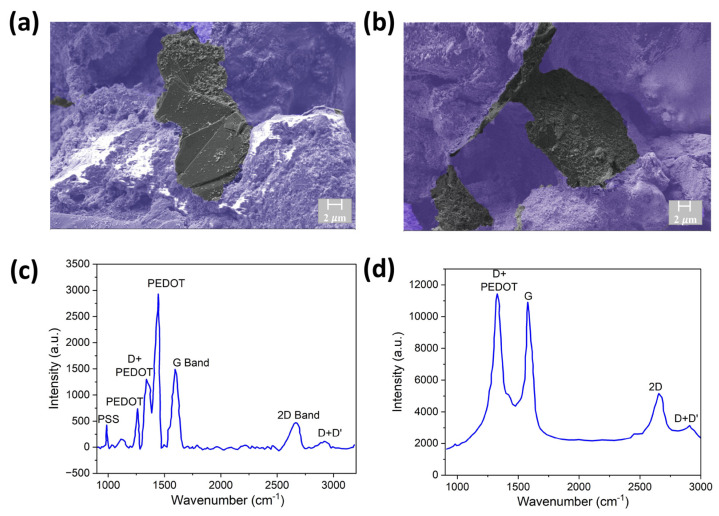
(**a**,**b**) Cross-sectional SEM images with false color showing graphene attached to the conductive ink and cellulose fibrils at different spatial locations (black indicates the graphene flakes, purple indicates the polymer network). Raman microscopy spectrum of the sensor (**c**) before oxidation and (**d**) after oxidation.

**Figure 3 sensors-23-08115-f003:**
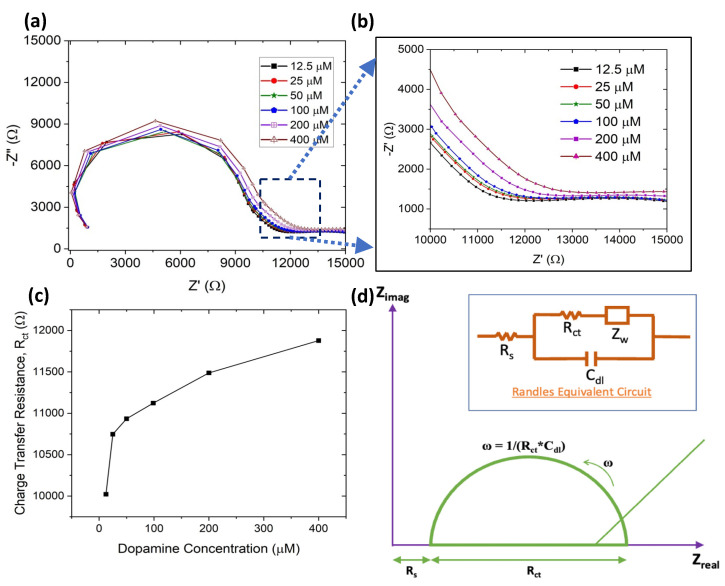
Single-layer dip-coated paper-based biosensor (**a**) EIS characterization of dopamine detection from 12.5 µM to 400 µM. (**b**) Enlarged view of the tail in (**a**,**c**) charge transfer resistance plot with dopamine concentration. (**d**) Randles equivalent circuit and its representative Nyquist plot.

**Figure 4 sensors-23-08115-f004:**
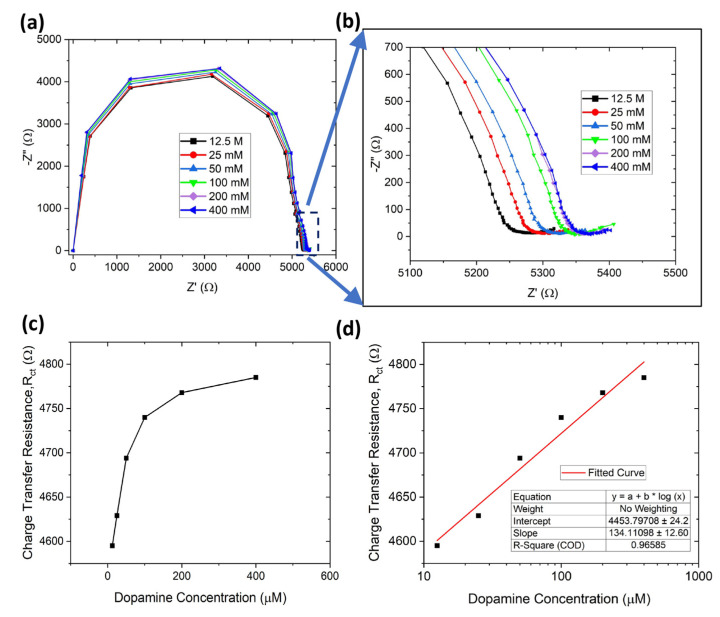
Double-layer dip-coated paper-based biosensor. (**a**) EIS characterization of dopamine detection from 12.5 µM to 400 µM, (**b**) enlarged view of (**a**,**c**) charge transfer resistance plot with dopamine concentration, (**d**) fitted linear plot for determining dopamine limit of detection.

**Figure 5 sensors-23-08115-f005:**
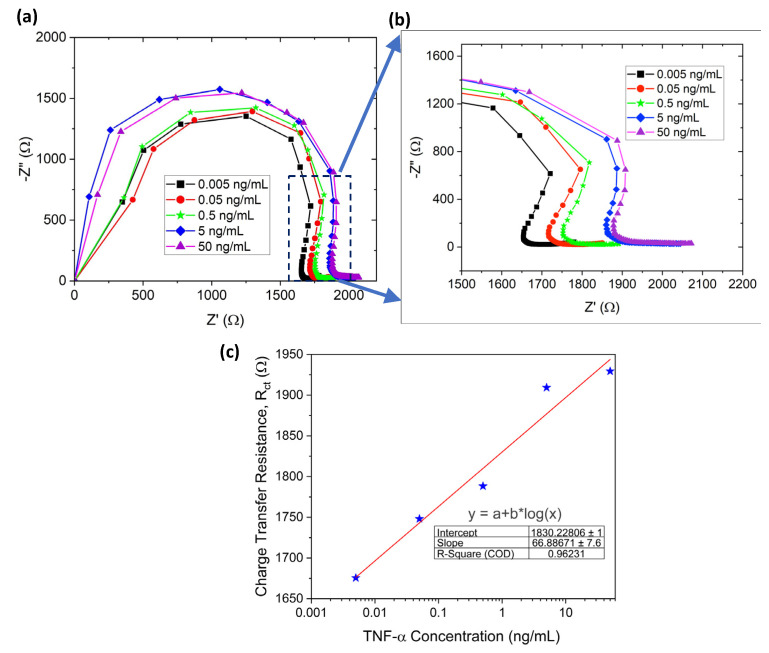
Paper-based biosensor. (**a**) EIS characterization of TNF-α detection from 0.005 ng/mL to 50 ng/mL, (**b**) enlarged view of (**a**,**c**) charge transfer resistance plot with the logarithmic scale of TNF-α concentration (straight red line depicts calibration curve).

**Figure 6 sensors-23-08115-f006:**
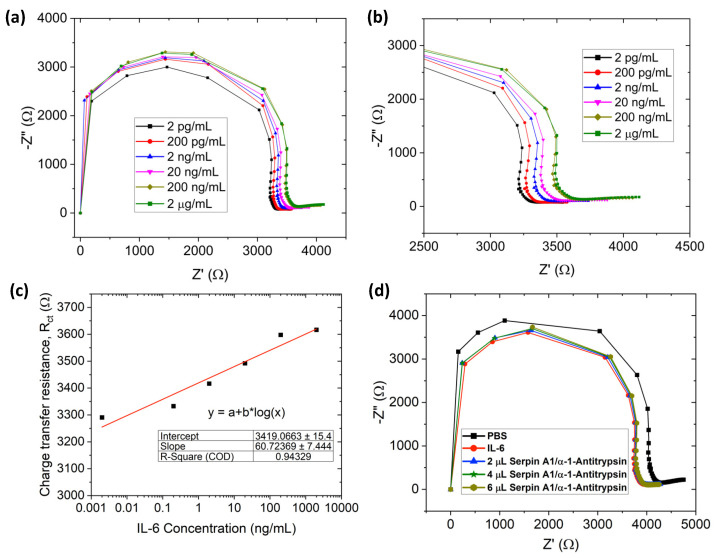
Paper-based biosensors. (**a**) EIS characterization of IL-6 detection from 0.002 ng/mL to 2000 ng/mL, (**b**) enlarged view of (**a**,**c**) charge transfer resistance plot with the logarithmic scale of IL-6 concentration (red straight line depicts calibration curve), (**d**) selective detection of IL-6.

**Figure 7 sensors-23-08115-f007:**
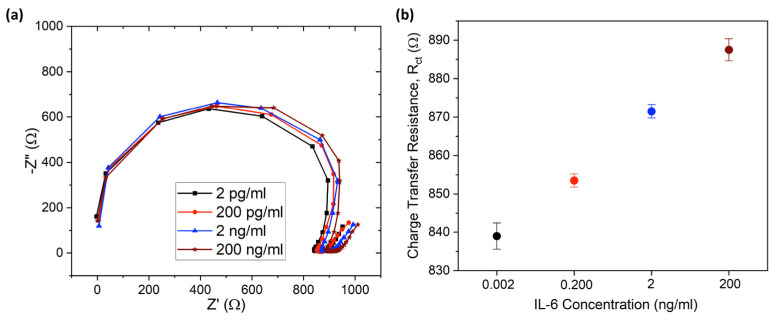
(**a**) EIS characterization of IL-6 detection in human serum from 2 pg/mL to 200 ng/mL, (**b**) charge transfer resistance with uncertainty plot for IL-6 concentration from 2 pg/mL to 200 ng/mL.

**Table 1 sensors-23-08115-t001:** Comparison of dopamine detection performance for different electrochemical biosensors based on conductive polymers.

Electrode Structure	Technique	Linear Range	Limit of Detection (LOD)	References
Polyaniline (PANI)–Au nanoparticles (NPs)	Differential pulse voltammetry (DPV)	10–1700 µM	5 µM	[77]
Polyaniline (PANI)–Au NPs	Linear sweep voltammetry (LSV)	20–100 μM	16 μM	[78]
PEDOT:PSS (FET)	CV, DPV	5–100 µM	6 µM	[79]
PEDOT: PSS/Chitosan/Graphene	CV, DPV	0.05–70 µM	0.29 µM	[80]
MWCNT-PEDOT	CV, DPV	10–330 µM	10 µM	[81]
Ti_3_C_2_Cl_2_/graphitic pencil electrode	CV, DPV, EIS	10−2000 μM	702 nM	[82]
Cu-benzene-1,3,5-tricarboxylic acid/carbon paste electrode	CV, DPV	0.05-500 μM	0.03 μM	[83]
Glassy carbon electrode (GCE)/Carbon quantum dots/CuO	Square Wave voltammetry (SWV)	1–800 μM	25.4 μM	[84]
GCE/Pt/Ti_3_C_2_T_x_	CV, Constant Voltage Deposition	50 nM–9 mM	50 nM	[85]
G-PEDOT:PSS	EIS	12.5–400 µM	3.4 µM	This work

**Table 2 sensors-23-08115-t002:** Comparison of TNF-α detection of some electrochemical biosensors using different techniques.

Biosensor Structure	Sensing Matrix	Technique	Detection Range	LOD	References
Poly(guanine)-functionalized silica NPs	Antibody	Square wave voltammograms	0.1–100 ng/mL	50 pg/mL	[86]
Alkaline phosphatase functionalized nanospheres	Antibody	EIS	0.02–200 ng/mL	0.01 ng/mL	[87]
Au working electrode	Aptamer	CV	10–100 ng/mL	10 ng/mL	[88]
Comb-structured Au microelectrode arrays	Antibody	EIS	0.001-1 ng/mL	1 pg/mL	[89]
MoS_2_ nanoflower	Antibody	CV & EIS	1-200 pg/mL	0.202 pg/mL	[90]
Au W.E.	Antibody	EIS	266–666,000 pg/mL	266 pg/mL	[91]
Si3N4/SiO2/Si[P]/Al	Antibody	Capacitive	1–30 pg/mL	1 pg/mL	[92]
G-PEDOT:PSS	Antibody	EIS	0.005–50 ng/mL	5.97 pg/mL	This work

## Data Availability

The data that support the findings of this study are available from the corresponding author upon reasonable request.

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
