# Peer review of "A Facile Graphene Conductive Polymer Paper Based Biosensor for Dopamine, TNF-α, and IL-6 Detection"

_sensors, 2023, doi:10.3390/s23198115_

Round 1

Reviewer 1 Report

The present manuscript entitled "A Facile Graphene Conductive Polymer Paper Based Biosensor for Dopamine, TNF-α, and IL-6 Detection" by Md Ashiqur Rahman, Ramendra Kishor Pal, Nazmul Islam, Robert Freeman, Francois Berthiaume, Aaron Mazzeo, and Ali Ashraf (sensors-2598240) is written correctly and has a good structure; moreover, it has all the necessary parts. The article is interesting from an analytical and sensor point of view; therefore, it should interest the reader. I proposed a few improvements. The paper meets Sensors' requirements, and I recommend the article for publication in Sensors following the common editing stage. My current decision is a minor revision. More specific comments and observations are presented below.

1. Keywords can be sorted alphabetically.

2. Page 3, line 101. Please explain EDC-NHS chemistry in this place.

3. Units should be standardized, e.g., pg mL-1 or pg/mL.

4. Please correct typos, e.g., indexes in units (page 3, line 115), unit “oC” (page 5, line 156), indexes in cations (page 6, line 182), and others.

5. Underdrawings are sometimes described in lowercase and sometimes in uppercase. It has to be unified.

6. Figure 2. You can also add a numeric description of the most important bands.

7. Page 5, line 160. Suddenly, reference [71] appears. Literature references should be listed in ascending order of numbers. Current literature references should be renumbered.

8. Section 2.6. Please add more information about measurements, e.g., acquisition time, laser power, number of measurements, averaging, and others.

9. Page 12. The "relationship" is mentioned. This term should be changed to "relation". The relationship tends to be used more broadly to describe the interactions between specific people or smaller groups of people.

10. Page 10, lines 377 and 378. There are no references like [102] and [103].

11. Does the conducted studies have disadvantages? Please add a short discussion.

12. Appropriate tools should be used to best characterize the developed sensor (e.g., RGB Additive Color Model to Analytical Method Evaluation). This may be of interest to the reader.

13. Conclusions. Please emphasize clearly the advantages of the research carried out.

14. References. Journal names are sometimes written in lowercase. This should be unified.

I hope that the comments presented will help improve the article.

Reviewer 2 Report

Here in this work, the authors have developed paper-based portable (point-of-care), affordable (laser cut) microfluidic biosensor for the detection of biomarkers in healthcare applications. The device is essentially a paper which is dip-coated (single/double layer) with conductive polymers functionalized graphene acting as an electrode.

The authors demonstrated electrochemical detection of biomolecules of interest such as dopamine, cytokine (TNF-a, IL-6) using CV and EIS and measured limit of detection (LOD) experimentally.

The authors showed proof of attachment between graphene conductive polymer with cellulose fiber using SEM and Raman spectroscopy. The authors have performed adequate morphological characterization for hierarchical structure of the sensing substrate, drying temperature dependent film stability, and ink conductivity, viscosity/ink thickness.

The methodology of analyte detection relies on increasing analyte concentration dependent increase in charge-transfer resistance, owing to increased biomolecule adsorption on the surface. Selective detection of antibody/antigen pair has been demonstrated. The dopamine, TNF-a detection performance for different electrochemical biosensors have been compare nicely. The ability of detection of IL-6 in human serum, and potential opportunity for early cancer detection is highly promising.

Comments:

1.     iIt would be better to use the same term to designate a given item. Such as ‘conductive polymers functionalized graphene’ and ‘graphene and conductive polymer’ have been used in different lines of the text.

2.     Line 180: The authors utilized a three-electrode setup instead of two-electrode for better quantification. It is often confusing to understand what are the electrodes. One can still identify, G-PEDOT:PSS as working electrode, the other two electrode in this three-electrode setup is not clearly mentioned in the text or figures. This should be done.

3.     In Figure 1b, the process of converting dip coated electrodes into the fabricated sensor is not clearly explained. Should be described elaborately.

4.     In Figure S1, the difference between graphene attached to the conductive polymer (S1b) and graphene with conductive polymer coating structure (S1c,d) is not clearly understood.

5.     The figures are presented very poorly. Particularly the data points. This can be vastly improved, which further enhances readability. Such as in Fig 3 or any other EIS measurement related data instead of just connecting the data points, a guide to eye line matching semicircle or semicircle fits would have been better. In fact, the authors argued that, the Nyquist plot obtained in Fig. 3a-b was fitted with Randles equivalent circuit 4 components: solution resistance (Rs), charge transfer resistance (Rct), double layer capacitance (Cdl), and Warburg diffusion (Zw)). So, the data and fit overlay should have been shown instead of connecting the data points only.

6.     In Fig. S5, the increase of negative peak current with increasing Dopamine concentration is evident. However, the linearity (if it is!!) is not shown. Should be plotted separately.

7.     For TNF-a detection, the biosensor was single or double-layer dip coated?

8.     The experiments lack demonstration of reproducibility of the measurements line duplication or triplication of same experiments. As a result, concentration dependent measurements missing error limit with respect to the measurements.

Reviewer 3 Report

This study describes A Facile Graphene Conductive Polymer Paper Based Biosensor for Dopamine, TNF-α, and IL-6 Detection. The concept is no doubt well-functioned. However, my main concern is the novelty and practicality of this paper sensor. Meanwhile, several issues also need to be addressed to clarify the overall value of the paper sensor.   

1. Overview and analysis of paper surface modification methods, comparison of advantages and disadvantages of polymer modification should be discussion in the paper, in introduction or in discussion section.

2. Will the water absorption of thick paper support affect the test results

3. As far as I know, activation of -COOH by NHS and EDC produces an intermediate product, which is easily hydrolyzed while activated. This process lasts for 8 hours. Is the activated product stable? Some literature suggests a reaction time of 0.5 hours, while others require an overnight reaction. How is this reaction time determined?

4. There is no error bar shown in all the tests. Did all the tests conduct once? If not, please add the test number and error bars to the contents and Figures. 

5. Please avoid terms such as “We used” and “we detect”. 

6. Are the electrode comparison materials in Table 1 and 2 all based on paper? If the performance of the sensor depends on the electrode structure and material, what are the advantages of paper-based biosensor in this work?

7. In Fig.1, there is a unified font and size. Some fonts are too small, while others are too large, which looks a bit strange.

8. The number in Fig. 3a overlaps with the brackets, please adjust.

9. The formula in Fig.5(d) is marked in black, while the formula in Fig.6(c) is marked in blue. Please unify.

10. Why use a line chart which marked black in Fig.6(c)?

11. the font in figures of the paper is too small to be clear, please adjust it.

12. Why is there no locally enlarged data in Fig.5? There is not data of selective detection of TNF-α.
13. What was the total analysis time of this proposed method?

Round 2

Reviewer 3 Report

Accept as it is, the revised manuscript has been satisfied the requirement of the journal.